# BayesPrompt: Prompting Large-Scale Pre-Trained Language Models on Few-shot Inference via Debiased Domain Abstraction

**Jiangmeng Li** [1 2*], **Fei Song** [1 3*], **Yifan Jin** [1 3], **Wenwen Qiang** [1†], **Changwen Zheng** [1 3]
**Fuchun Sun** [1 4], **Hui Xiong** [5]

[1]National Key Laboratory of Space Integrated Information System, Institute of Software Chinese Academy of Sciences, Beijing, China
[2]State Key Laboratory of Intelligent Game, Beijing, China
[3]University of Chinese Academy of Sciences, Beijing, China
[4]Tsinghua University, Beijing, China
[5]The Hong Kong University of Science and Technology (Guangzhou), Guangzhou, China
`{jiangmeng2019,songfei2022,yifan2020,qiangwenwen}@iscas.ac.cn`
`changwen@iscas.ac.cn,fcsun@mail.tsinghua.edu.cn,xionghui@ust.hk`

## Abstract

As a novel and effective fine-tuning paradigm based on large-scale pre-trained language models (PLMs), prompt-tuning aims to reduce the gap between downstream tasks and pre-training objectives. While prompt-tuning has yielded continuous advancements in various tasks, such an approach still remains a persistent defect: prompt-tuning methods fail to generalize to specific few-shot patterns. From the perspective of distribution analyses, we disclose that the intrinsic issues behind the phenomenon are the *over-multitudinous* conceptual knowledge contained in PLMs and the *incomplete* knowledge for target downstream domains, which jointly result in that PLMs *mis-locate* the knowledge distributions corresponding to the target domains in the universal knowledge embedding space. To this end, we intuitively explore to approximate the complete target domains of downstream tasks in a debiased manner, and then abstract such domains to generate discriminative prompts, thereby providing the de-ambiguous guidance for PLMs. Guided by such an intuition, we propose a simple yet effective approach, namely *BayesPrompt*, to learn prompts that contain the domain discriminative information against the interference from domain-irrelevant knowledge. BayesPrompt primitively leverages known distributions to approximate the debiased factual distributions of target domains and further uniformly samples certain representative features from the approximated distributions to generate the ultimate prompts for PLMs. We provide theoretical insights with the connection to domain adaptation. Empirically, our method achieves state-of-the-art performance on benchmarks[1].

## 1 Introduction

Benefiting from key ingredients of the massive candidate datasets, vast trainable model parameters, and choreographed training architecture, PLMs (Dai & Le, 2015), as artificial general intelligence approaches, achieve impressive successes in general natural language processing fields. However, for specialized downstream tasks, PLMs hit a developmental bottleneck, which especially falls short of the expectations of researchers in few-shot scenarios (Wang et al., 2020). The intrinsic reason behind such an issue is that PLMs contain ***over-multitudinous* conceptual knowledge**[2], resulting in that domain-irrelevant knowledge may interfere with the inference on downstream tasks, especially for

---

*Equal contribution.
†Corresponding author.

[1]The code implementation of our method is available at this link.
[2]The knowledge contained by PLMs exhibits inherent polysemy.

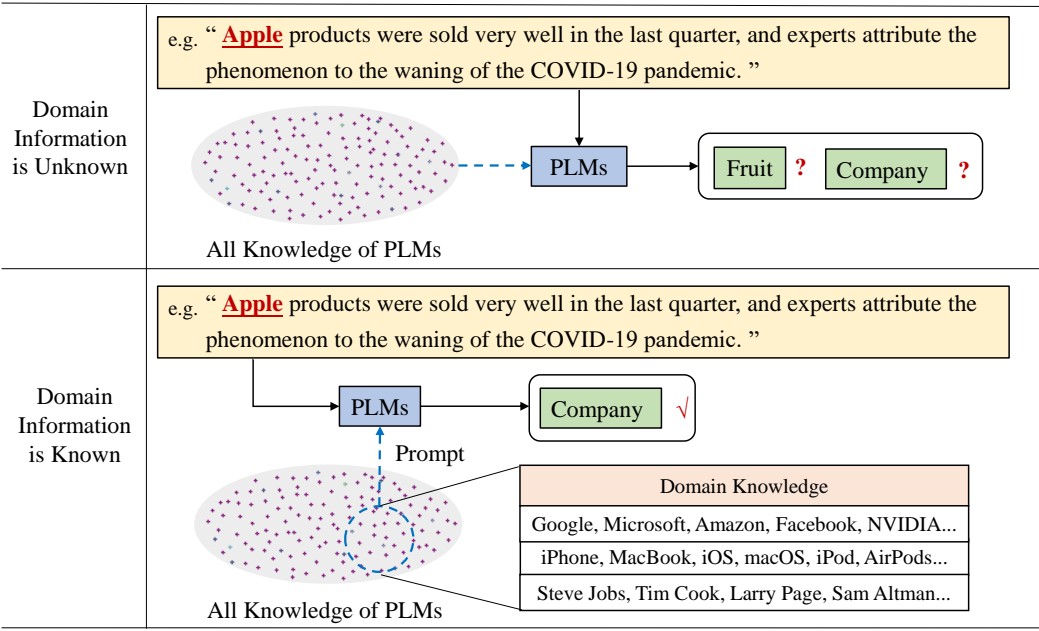

Figure 1: Examples of leveraging the domain information prompting PLMs. The points within the blue dashed circle represent the factual knowledge distribution of task-related data in PLMs.

few-shot datasets (Li et al., 2023). We demonstrate an illustrative example in Figure 1 for ease of understanding.

To remedy this deficiency, recent works propose well-designed prompts to guide PLMs thereby avoiding the inference outliers on downstream tasks (Wei et al., 2021a; Liu et al., 2023b). However, manually constructing such prompts requires expertise and costly workloads (Schick & Schütze, 2020; Li et al., 2022b). To this end, the data-driven trainable prompts emerge and yield significant performance boosts on the downstream inference of PLMs (Gao et al., 2021; Liu et al., 2021; Gu et al., 2021), but such a learning paradigm of prompt still suffers from a long-standing challenge: the limited and discrete semantic information contained in the training samples from downstream domains can barely support the conventional trainable prompts to acquire sufficient supervision, such that the guidance of the generated prompts is *trivial* to PLMs. Especially, such a challenge further exacerbates the performance of PLMs in few-shot scenarios.

To further understand the implicit and essential reason behind the defect of PLMs in the few-shot scenarios, we revisit the operational principle introduced in downstream inferences of PLMs from the distribution perspective. For the traditional inference paradigm without prompts demonstrated in Figure 2 (a), certain samples, e.g., sentences, may contain information that directly confounds the inference of PLMs. We ascribe this phenomenon to the fact that the confounding samples concurrently belong to *multiple* domain distributions in the knowledge embedding space of PLMs, and the model cannot determine the desired domain without prompts that contain **domain discriminative information**[3]. Consequently, the over-multitudinous conceptual knowledge can not only *empower* PLMs to understand universal concepts but also *interfere* with the inference on specific tasks. For the inference paradigms with trainable prompts demonstrated in Figure 2 (b) and (c), the information contained in the *limited* training samples of candidate downstream domains may lead to the knowledge ambiguity of PLMs, while the information contained in the corresponding *complete* domains can effectively cope with such an issue. We conjecture that the limited training samples lead the trainable prompts to learn the biased empirical distribution of the target domain, which only contains partial information and is inconsistent with the factual distribution of the target domain, resulting in the covariate shift problem (Heckman, 1979; Shimodaira, 2000), thereby still providing certain ambiguous guidance for PLMs.

---

[3]The information that effectively characterizes features of the actual downstream domain.

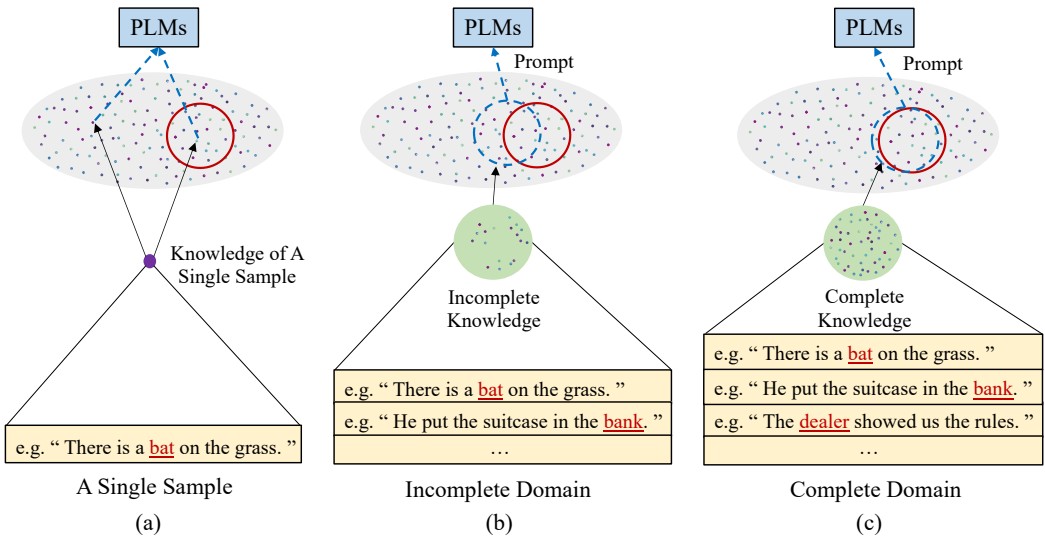

Figure 2: The intrinsic reasons why patchy target domain knowledge may incur negative impacts on the inference of PLMs. The points in the red solid circles adhere to the factual distributions corresponding to the task-related data in PLMs, while the points in the blue dashed circles represent the approximated domain distributions obtained through a specific learning paradigm.

To this end, we intuitively explore to approximate the complete training domains on downstream tasks in a debiased manner, and then abstract such domains to generate discriminative prompts, thereby providing the **de-ambiguous guidance**[4] for PLMs. Specifically, we propose a novel approach, dubbed *BayesPrompt*, which primitively leverages known distributions, e.g., Gaussian distribution, to approximate the **debiased factual distributions**[5] of downstream domains and further uniformly samples certain representative features from the approximated distributions to generate the ultimate prompts for PLMs. On top of this, the behavior of BayesPrompt can be treated as the **debiased domain abstraction**[6]. We elaborate on the procedures of the proposed approach in a nutshell. The distribution approximation is achieved by using Stein Variational Gradient Descent (SVGD) (Liu & Wang, 2016), which is a general-purpose Bayesian inference algorithm. In practice, we observe that selecting the conventional Gaussian distribution as the known distribution degenerates the approximation of downstream domain distributions, and thus the Gaussian Mixture Model (GMM) (Reynolds et al., 2009) is constructed to fit the sample distribution. The resulting distribution and sample representations are then used to initialize the target distribution and particles for the SVGD algorithm. Through iterative updates of SVGD, a new set of particles is generated that approximates the target distribution. As for the sampling strategy, we adopt uniform sampling based on empirical evidence. By sampling from the target distribution, we obtain prompts that contain domain discriminative information, which can mitigate interference from domain-irrelevant knowledge. The sufficient experimental analyses prove that BayesPrompt achieves state-of-the-art performance. **Contributions**:

- We disclose a long-standing issue challenging the downstream inference of PLMs for current methods, especially in few-shot scenarios, which is further described by providing intuitive illustrations for ease of understanding.

- We propose BayesPrompt, orthogonal to existing methods, to approximate the factual distributions of downstream domains in a debiased manner, and further abstract such domains thereby generating discriminative prompts for PLMs.

- We conduct extensive evaluations on various experimental settings, including the few-shot experiments and standard experiments, to empirically prove the effectiveness of BayesPrompt.

---

[4]Using a feature that well describes the actual downstream domain as a prompt to guide the PLMs to remember the knowledge of the target domain.

[5]A distribution that fits the actual downstream domain distribution well.

[6]Abstracting a feature that can well describe the actual downstream domain based on the debiased factual distribution.

## 2 RELATED WORKS

PLMs have demonstrated their ability to capture rich knowledge from massive corpora (Li et al., 2022c). However, there exists a significant gap between the objectives of pre-training and fine-tuning. Prompt-tuning methods are fueled by the birth of GPT-3 (Brown et al., 2020) and have achieved impressive successes in a wide range of tasks. By leveraging language prompts as contexts, downstream tasks can be expressed as certain objectives similar to pre-training objectives. With appropriate manual prompts, a series of studies (Ben-David et al., 2021; Lester et al., 2021; Lu et al., 2021; Jiang et al., 2022; Ma et al., 2022) have been proposed, demonstrating the advancement of prompt-tuning. (Han et al., 2022) propose a model, called PTR, which creatively applies logic rules to construct prompts with several sub-prompts. (Ding et al., 2021) apply prompt-tuning to entity typing by constructing an entity-oriented verbalizer and templates. (Hu et al., 2021) incorporate external knowledge into the verbalizer with calibration. (Li et al., 2022a) use external knowledge bases to construct knowledge-injected prompts. (Chen et al., 2022b) inject latent knowledge contained in labels into prompt construction and synergistically optimize their representation with structured constraints. (Wang et al., 2023) exploit knowledge to improve the efficiency of text classification. (Liu et al., 2023a) leverage prompt-tuning to incorporate background information and relational information for causal reasoning. These previous studies have established that the effectiveness of prompt-based learning is largely attributed to the implicit knowledge embedded in PLMs. However, they did not focus on the factual distributions of target domains in PLMs. Recently, advancements in large language models have enabled the emergence of various forms of prompt-tuning (Brown et al., 2020; Wei et al., 2021b; 2022). However, the performance improvements emerge only with a sufficient model scale. In this paper, we propose BayesPrompt to attenuate the impact of domain-irrelevant knowledge while enhancing the adaptability to different model scales. As compared to the above-mentioned prompt-tuning methods, BayesPrompt achieves a good balance among model effectiveness, model generalization, model scale, and human workloads.

## 3 PRELIMINARY

Before introducing BayesPrompt, we take Relation Extraction (RE) as an example to recap the preliminaries of prompt-tuning. Formally, we suppose the RE dataset as $D = \{X, Y\}$, where $X$ denotes the set of examples, and $Y$ denotes the set of relation labels. To be concise, we use $w_s$ and $w_o$ to represent all entities briefly, as a single entity may be composed of multiple tokens. Therefore, each example $x \in X$ consists of several tokens, $x = \{w_1, w_2, w_s, ..., w_o, ..., w_n\}$. The RE task takes a query sentence $x$ with the corresponding entity pair $(w_s, w_o)$ as the input, aiming to learn a distribution $\mathcal{P}(y|(w_s, w_o))$ over all possible pre-defined relations $y \in Y$.

A typical prompt consists of a template $T(\cdot)$ and a set of label words $V$, where the template $T(\cdot)$ defines the location and number of the added auxiliary words, and $V$ refers to a set of label words in the vocabulary of a Language Model (LM). In addition to retaining the original tokens in $x$, one or more $[MASK]$ is placed into $x_{prompt}$ for the LM to fill the label words. Given each example $x$ for RE, the template $T(\cdot)$ is leveraged to insert pieces of texts with entity pair $(w_s, w_o)$ into $x$ to map $x$ as $x_{prompt} = T(x)$, where the $x_{prompt}$ is the corresponding input of LM with a $[MASK]$ token in it. Inspired by (Shin et al., 2020; Li & Liang, 2021; Liu et al., 2021; Zhou et al., 2022), we suppose the bijective mapping from the relation label space to the label word space as $M : Y \rightarrow V$, and $M(y)$ presents the label words corresponding to label $y$. Thus, the probability distribution over $V$ at the masked position can be implemented by

$$\mathcal{P}(y \mid x) = \mathcal{P}([MASK] = M(y) \mid T(x)). \tag{1}$$

By filling $[MASK]$ tokens in the input, the RE task can be transformed into a masked language modeling problem, where the goal is to infer the appropriate label words at the masked positions.

## 4 METHODOLOGY

Our objective is to learn prompts that contain the domain discriminative information against the interference from the domain-irrelevant knowledge by approximating the debiased factual distributions of downstream domains. To this end, we propose BayesPrompt to determine the discriminative

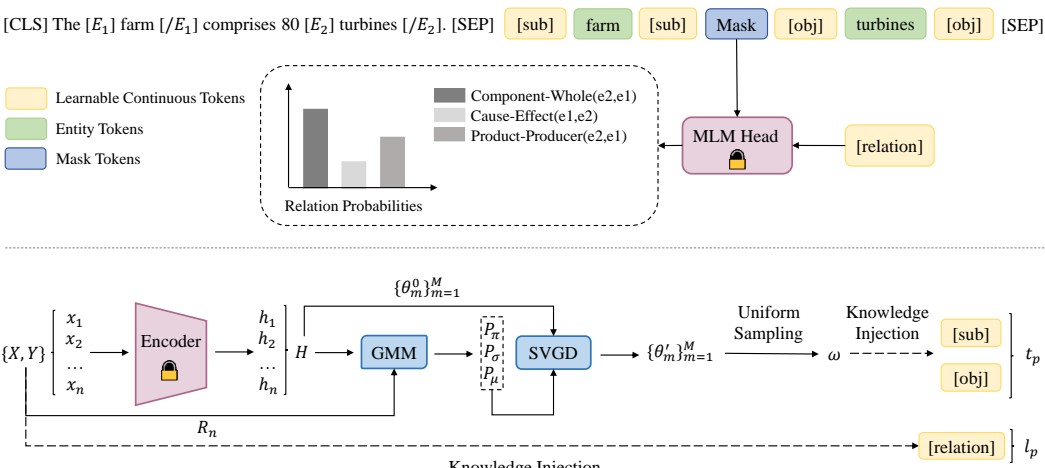

Figure 3: The overall framework of BayesPrompt. "[E]" and "[/E]" denote the start and end tags to indicate entities for inferring relations, respectively.

prompts. BayesPrompt provides unambiguous guidance for PLMs by debiasing training domains on downstream tasks and abstracting such domains in a uniform sampling manner.

The overall framework of BayesPrompt is illustrated in Figure 3. Specifically, the example $x_i$ is first fed into the encoder to obtain its representation $h_i$. As a practical observation discussed in **Section** 6, we notice that using the conventional Gaussian distribution as the known distribution degenerates the approximation of downstream domain distributions. Therefore, we build a GMM to model the representation distribution as follows:

$$P_\mu, P_\sigma, P_\pi = GMM\left(H, R_n\right), \tag{2}$$

where $H$ denotes the representations of all examples, and $R_n$ is the number of Gaussian components that are determined by relation categories. The parameters of each Gaussian component in the fitted GMM are represented by the outputs $P_\mu, P_\sigma$, and $P_\pi$. They denote the mean vectors, covariance matrices, and weights of each Gaussian component, respectively.

We adopt SVGD (Liu & Wang, 2016) to approximate the debiased factual distributions of downstream domains. SVGD is a general-purpose variational inference algorithm that aims to approximate a target distribution. It converges more rapidly than MCMC (Hastings, 1970) because it utilizes the gradient of the target distribution and follows a deterministic update principle. In our work, the Gaussian mixture distribution determined by $P_\mu, P_\sigma$, and $P_\pi$ is fed into SVGD as the target distribution, and the representations of training examples are considered as the set of initial particles $\Theta = \left\{\theta_m^0\right\}_{m=1}^M$, where $M$ is equal to the number of examples. By applying a form of functional gradient descent that minimizes the KL divergence (Kullback & Leibler, 1951), SVGD iteratively transports the set of initial particles to match the target distribution. At the iteration $\ell$, the particle $\theta_m \in \Theta$ is updated by

Table 1: Disassembled relations prepared for label prompt words.

| Relation Labels | $S_r$ |
| --- | --- |
| Component-Whole(e1,e2) | {"Component", "Whole"} |
| Message-Topic(e1,e2) | {"Message", "Topic"} |
| Cause-Effect(e1,e2) | {"Cause", "Effect"} |
| Instrument-Agency(e1,e2) | {"Instrument", "Agency"} |
| Content-Container(e1,e2) | {"Content", "Container"} |
| Product-Producer(e1,e2) | {"Product", "Producer"} |
| Member-Collection(e1,e2) | {"Member", "Collection"} |

$$\theta_m^{\ell+1} \leftarrow \theta_m^\ell + \epsilon_\ell \phi\left(\theta_m^\ell\right), \text{ where } \phi\left(\theta\right) = \frac{1}{M}\sum_{j=1}^M\left[k\left(\theta_j^\ell, \theta\right)\nabla_{\theta_j^\ell}\log p\left(\theta_j^\ell\right) + \nabla_{\theta_j^\ell}k\left(\theta_j^\ell, \theta\right)\right], \tag{3}$$

where $\epsilon_\ell$ is the step-size at the $\ell$-th iteration, $k$ denotes the RBF kernel, and $p_\theta$ represents the target distribution determined by GMM. A particle obtains information from other particles by requesting their gradients, and further determines its own update direction. The relevance of other

particles is evaluated based on the kernel distance, thereby assigning larger weights to particles that are closer (Yoon et al., 2018). The last term $\nabla_{\theta_j^\ell} k\left(\theta_j^\ell, \theta\right)$ employs the repulsive force between particles to prevent them from collapsing into a trivial point. Through iterative updates, the resulting particles $\Theta' = \left\{\theta_m'\right\}_{m=1}^M$ approximate the debiased factual distributions of downstream domains. By uniform sampling from $\Theta'$, we can obtain the latent knowledge $\omega$, which represents a debiased domain abstraction that provides de-ambiguous guidance for PLMs. Subsequently, the prompt with knowledge injection can be constructed for the RE task.

To fully leverage the substantial semantic knowledge embedded in relation labels, we define a label prompt word $l_p$ to represent the implicit semantics of the relation. Inspired by (Chen et al., 2022b), we obtain a set of semantic words $S_r$ by disassembling the relation label $r$ and set $\mathrm{S}_R = [\mathrm{S}_{r1}, \mathrm{S}_{r2}, \ldots, \mathrm{S}_{rm}]$, where $m$ is the number of relation labels. The specific disassembly process is shown in Table 1. We suppose the probability distribution over the semantic words sets $S_R$ as $\phi_R = [\phi_{r1}, \phi_{r2}, \ldots, \phi_{rm}]$, where the probability distribution is estimated through frequency statistics. Furthermore, we impose the weighted average function for $\phi_r$ on each word among $S_r$ to initialize the label prompt word $l_p$, which can inject the semantic knowledge of relations as follows:

$$\hat{e}\left(l_p\right) = \phi_r \cdot e\left(S_r\right), \tag{4}$$

where $\hat{e}\left(l_p\right)$ is the embedding of the label prompt word $l_p$, and $e$ represents the word-embedding layer of LM. By injecting the semantic knowledge about the label into the label prompt word $l_p$, we can facilitate the process of inferring relations.

Note that the Type Marker (Zhou & Chen, 2021) approach can enhance performance by incorporating the type information of entities, but it requires additional annotations for entity types that are commonly unavailable. Inspired by the label prompt word $l_p$, we further construct the type prompt word $t_p$ to inject entity type information into prompts. For the relation label "per:date_of_birth", it is evident that the subject entity matching such a relation belongs to "person", and the object entity matching such a relation belongs to "date". With the prior knowledge contained in a specific relation, we can intuitively acquire the scope of potential entity types, instead of relying on annotations. However, the applicability of this method is limited. For instance, given the relation "Cause-Effect(e1,e2)", we cannot estimate the potential entity types, since this relation does not provide any information on the entities involved. To remedy this deficiency, we initialize the type prompt word $t_p$ with the latent knowledge $\omega$ obtained by uniform sampling:

$$\hat{e}\left(t_p\right) = \omega, \tag{5}$$

where $\hat{e}\left(t_p\right)$ is the embedding of type prompt word $t_p$. The improved performance shows the effectiveness of the discriminative information contained in latent knowledge $\omega$.

---

**Algorithm 1** BayesPrompt

**Input:** A set of training samples $\{x_i, y_i\}_{i=1}^n$.
\# Generating features by using a fixed PLM encoder $E\left(\cdot\right)$
$h_i = E(x_i)$, and $H = \{h_i\}_{i=1}^n$
\# Initializing a GMM to model the representation distribution
$R_n = |y|$, and $P_\mu, P_\sigma, P_\pi = \mathrm{GMM}\left(H, R_n\right)$
**Initialized particles:** $\left\{\theta_m^0\right\}_{m=1}^M = H$.
\# Adopting SVGD to approximate the debiased factual distribution
**for** iteration $\ell$ **do**
 $\theta_m^{\ell+1} \leftarrow \theta_m^\ell + \epsilon_\ell \phi\left(\theta_m^\ell\right)$, where $\phi\left(\theta\right) = \frac{1}{M}$
 $\sum_{j=1}^M \left[k\left(\theta_j^\ell, \theta\right) \nabla_{\theta_j^\ell} \log p\left(\theta_j^\ell\right) + \nabla_{\theta_j^\ell} k\left(\theta_j^\ell, \theta\right)\right]$
**end for**
**Updated particles:** $\Theta' = \left\{\theta_m'\right\}_{m=1}^M$.
**for** $t$-th training iteration **do**
 \# Sampling to initialize prompts
 $\omega \sim \Theta'$, and $\hat{e}\left(t_p\right) = \omega$
 $\hat{e}\left(l_p\right) = \phi_r \cdot e\left(S_r\right)$
 \# Minimize the loss to train the RE task
 $min\left\{\mathcal{J} = -\frac{1}{|X|} \sum_{x \in X} y log \mathcal{P}\left(y \mid x\right)\right\}$
**end for**

---

To fully associate the initialized label prompt word $l_p$ and type prompt word $t_p$ with the surrounding context, we perform further optimization of their representation by the loss function computed as the cross-entropy between $y$ and $\mathcal{P}\left(y \mid x\right)$ as follows:

$$\mathcal{J} = -\frac{1}{|X|} \sum_{x \in X} y log \mathcal{P}\left([MASK] = M\left(y\right) \mid T\left(x\right)\right), \tag{6}$$

where $|X|$ represents the number of samples in the training dataset. The procedure of BayesPrompt is summarized in Algorithm 1.

Table 2: F1 scores (%) of prompt-tuning models with different settings. The best results are in **bold**. $\triangle$(B-K) denotes the comparison between BayesPrompt and KnowPrompt, and $\triangle$(B-R) denotes the comparison between BayesPrompt and RetrievalRE.

| | | Few-Shot Setting | | | | | | | |
|---|---|---|---|---|---|---|---|---|---|
| Datasets | Split | FINE-TUNING | GDPNET | PTR | KnowPrompt | RetrievalRE | **BayesPrompt** | $\triangle$(B-K) | $\triangle$(B-R) |
| SemEval | K=1 | 18.5(±1.4) | 10.3(±2.5) | 14.7(±1.1) | 28.6(±6.2) | 33.3(±1.6) | **35.1**(±2.9) | | |
| | K=5 | 41.5(±2.3) | 42.7(±2.0) | 53.9(±1.9) | 66.1(±8.6) | 69.7(±1.7) | **71.6**(±3.3) | **+4.3** | **+1.23** |
| | K=16 | 66.1(±0.4) | 67.5(±0.8) | 80.6(±1.2) | 80.9(±1.6) | **81.8**(±1.0) | **81.8**(±1.2) | | |
| TACRED | K=1 | 7.6(±3.0) | 4.2(±3.8) | 8.6(±2.5) | 17.6(±1.8) | 19.5(±1.5) | **22.5**(±2.5) | | |
| | K=5 | 16.6(±2.1) | 15.5(±2.3) | 24.9(±3.1) | 28.8(±2.0) | 30.7(±1.7) | **31.4**(±0.6) | **+3** | **+1.27** |
| | K=16 | 26.8(±1.8) | 28(±1.8) | 30.7(±2.0) | 34.7(±1.8) | 36.1(±1.2) | **36.2**(±0.8) | | |
| TACREV | K=1 | 7.2(±1.4) | 5.1(±2.4) | 9.4(±0.7) | 17.8(±2.2) | 18.7(±1.8) | **21.9**(±2.0) | | |
| | K=5 | 16.3(±2.1) | 17.8(±2.4) | 26.9(±1.5) | 30.4(±0.5) | 30.6(±0.2) | **31.2**(±0.8) | **+2.43** | **+1.37** |
| | K=16 | 25.8(±1.2) | 26.4(±1.2) | 31.4(±0.3) | 33.2(±1.4) | 35.3(±0.3) | **35.6**(±0.7) | | |

## 5 THEORETICAL INSIGHTS WITH CONNECTION TO DOMAIN ADAPTATION

**The gap between prompting problem and domain adaptation**. The "domain adaptation (Qiang et al., 2021)" is learning from a source data distribution a well-performing model on a different (but related) target data distribution, while, such a purpose has a gap with our purpose in BayesPrompt. Our method aims to fit the distribution of a few-shot domain, but we are not going to align the distributions of the target few-shot domain and the domain of PLMs. The intuition behind such a behavior is that the distribution of the PLM domain is subject to the Gaussian distribution but the distribution of the few-shot domain is not the Gaussian distribution, such that arbitrarily aligning the distributions to fine-tune the PLM degenerates its ability to capture discriminative information, which is also empirically proved by (Kumar et al., 2022).

The reason for such a statement is that the innate assumption behind the Gaussian distribution is the sufficient samples, i.e., when the statistical samples are sufficient, the Gaussian distribution well fits the factual distribution of such discrete samples. However, for the few-shot scenario, the data is limited, such that the distribution can not be well fit by the Gaussian distribution. Therefore, if we directly fine-tune the PLM by aligning the distribution of the PLM domain (s.t. Gaussian distribution) with the distribution of the few-shot domain (s.t. another distribution), the knowledge of PLMs can be perturbed, and further, the discriminative information cannot be learned by PLMs.

**Do the theoretical assumptions on a shared label space from domain adaptation hold in prompt-tuning?** The behavior of BayesPrompt can be treated as implicitly adapting the target downstream domain to a subset of a specific PLM domain by leveraging a well-learned prompt. We will discuss the feasibility of transforming the theoretical objective of BayesPrompt to that of domain adaptation and why the domain adaptation cannot work in this scenario.

For "*the feasibility of the transformation*", in the prompt-tuning scenario, the downstream domain can be treated as the target domain, and the specific subset of the PLM domain can be treated as the source domain, i.e., the domain distribution alignment is performed between the specific subset of the PLM domain and the downstream domain, which have the shared labels. Furthermore, due to the mechanism behind prompt-tuning, i.e., the PLM network is frozen, we determine that both the pre-training data and the data from the downstream dataset are fed into the shared network to be projected into the shared latent space, and there is no any extra non-linear projection during the prediction. The only difference in features from the pre-training domain and the downstream domain is the data, such that we derive a conclusion that the specific subset of the PLM domain and the downstream domain have a shared label space, and the analyses based on the assumption of domain adaptation are theoretically feasible.

For "*the reason why domain adaptation cannot work in the proposed scenario*", following the aforementioned analyses, we disclose that the downstream domain can be bounded by the discrete data, but the specific subset of the PLM domain, which has the shared labels with the downstream domain, cannot be certainly determined, such that conventional domain adaptation methods cannot be directly leveraged to achieve the objective.

Table 3: Standard RE performance of F1 scores (%) on benchmarks. "w/o" indicates that the model does not use additional data, yet "w/" indicates the model requires extra data for downstream tasks.

| Methods | Extra Data | SemEval | TACRED | TACREV | RE-TACRED | Average |
|---|---|---|---|---|---|---|
| | | Standard Setting | | | | |
| | | Fine-tuning pre-trained models | | | | |
| FINE-TUNING | w/o | 87.6 | 68.7 | 76.0 | 84.9 | 79.3 |
| SPANBERT | w/ | - | 70.8 | 78.0 | 85.3 | 78.0 |
| KNOWBERT | w/ | 89.1 | 71.5 | 79.3 | 89.1 | 82.3 |
| LUKE | w/ | - | 72.7 | 80.6 | - | 76.7 |
| MTB | w/ | 89.5 | 70.1 | - | - | 79.8 |
| GDPNET | w/o | - | 71.5 | 79.3 | - | 75.4 |
| | | Prompt-tuning pre-trained models | | | | |
| PTR | w/o | 89.9 | 72.4 | 81.4 | 90.9 | 83.7 |
| KnowPrompt | w/o | 90.2 | 72.4 | 82.4 | 91.3 | 84.1 |
| RetrievalRE | w/o | 90.4 | 72.7 | 82.7 | **91.5** | 84.3 |
| **BayesPrompt** | w/o | **90.6** | **72.9** | **83.0** | 91.4 | **84.5** |

## 6 EXPERIMENTS

**Benchmarking BayesPrompt on various datasets**. We evaluate BayesPrompt on four RE datasets, namely SemEval 2010 Task 8 (SemEval) (Hendrickx et al., 2019), TACRED (Zhang et al., 2017), TACREV (Alt et al., 2020), and ReTACRED (Stoica et al., 2021). We adopt the F1 score as the primary evaluation metric for the experiments on all the aforementioned datasets. In experiments, we select KnowPrompt (Chen et al., 2022b) as our primary baseline, which is a representative model that injects the latent knowledge contained in labels into the prompt construction, thereby empowering the inference of relations. We further compare BayesPrompt with RetrievalRE (Chen et al., 2022a), which is the follow-up work of KnowPrompt. In the few-shot learning setting, we perform 1-, 5-, and 16-shot experiments to assess the effectiveness of our method in low-resource scenarios. Following the benchmark experimental settings (Chen et al., 2022a), we report the F1 score and the standard deviation on different benchmark datasets in Table 2. The results demonstrate that on average, BayesPrompt beats KnowPrompt by 3.24% among benchmark datasets. For RetrievalRE, BayesPrompt achieves an average improvement of 1.29% among benchmark datasets. The statistical results further demonstrate the effectiveness of BayesPrompt. Besides, we also perform the significance test, i.e., t-test, with the primary baseline, and observe that the P values are consistently lower than 0.05, e.g., 0.045 on the SemEval dataset, indicating that the improvement of BayesPrompt is significant.

**Validation of the robustness of BayesPrompt against knowledge ambiguity**. We perform further experiments to prove that BayesPrompt is able to mitigate the negative impact of knowledge ambiguity, and the results are shown in Table 3. Considering the importance of entity knowledge in facilitating the understanding of relational semantics for models, we select SpanBERT (Joshi et al., 2020), KnowBERT (Peters et al., 2019), LUKE (Yamada et al., 2020), and MTB (Soares et al., 2019) as baselines, which are representative models leveraging the external knowledge to enhance learning objectives, input features, model architectures, or pre-training strategies. As shown in Table 3, BayesPrompt outperforms the compared knowledge-enhanced models, indicating that despite the task-specific knowledge already contained in knowledge-enhanced PLMs, it remains challenging for fine-tuning to sufficiently leverage such knowledge for downstream tasks. We compare BayesPrompt with the principal baseline, i.e., KnowPrompt, in the standard setting, and the results demonstrate that BayesPrompt can yield a 0.4% performance rise on average. When compared to the RetrievalRE, BayesPrompt still achieves an average performance improvement of 0.2%, further highlighting its superiority. The intriguing outcome demonstrates the merit of the proposed method: even when the amount of data changes, BayesPrompt remains robust in alleviating the negative impact of the knowledge ambiguity, i.e., the model maintains consistent performance.

**Training Complexity of BayesPrompt**. We compare the training complexity of BayesPrompt and KnowPrompt on the SemEval and TACREV datasets through experiments. The results are presented in Table 4, which indicates that BayesPrompt has a slightly higher time complexity than the benchmark method due to the necessity of adding prompts containing domain discriminative information for downstream tasks during each prediction to provide de-ambiguous guidance for PLMs. Nevertheless, the results in Table 2 and Table 3 demonstrate that although BayesPrompt has

Table 4: The complexity comparisons between BayesPrompt and KnowPrompt on SemEval and TACREV. Note that for fair comparisons, this experiment is based on 1 GPU of NVIDIA 3090.

| Methods | Parameters | Datasets | Training time cost for an epoch | | | |
|---------|-----------|----------|--------|--------|---------|-------------|
| | | | 1-shot | 5-shot | 16-shot | full dataset |
| KnowPrompt | 355M | SemEval | 16s | 19s | 26s | 198s |
| | | TACREV | 217s | 228s | 239s | 2096s |
| BayesPrompt | 355M | SemEval | 27s | 31s | 43s | 242s |
| | | TACREV | 332s | 344s | 376s | 2422s |

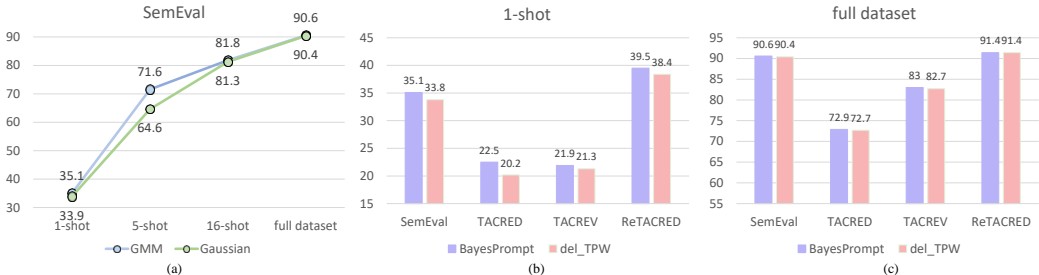

Figure 4: (a) GMM vs. Gaussian results on the SemEval dataset regarding different shots. (b) and (c) the ablation study on different datasets, where del_TPW refers to the removal of type prompt words.

a slightly higher time complexity, its performance improvement is consistent and substantial. For instance, in the 1-shot setting of the SemEval dataset, the training time cost of each epoch at the main baseline is 16 seconds, while the training time cost of each epoch at BayesPrompt is 27 seconds. However, the main baseline achieves an F1 score of 28.6%, whereas BayesPrompt achieves an F1 score of 35.1%.

**Ablation study**. For the approximation of debiased factual distributions, we comprehensively consider the Gaussian distribution and GMM as the candidate *known distributions*. The empirical results are demonstrated in Figure 4 (a), and we observe that our method using GMM achieves relatively suitable and effective performance. Figure 4 (b) and (c) demonstrate the effects of the discriminative prompts. Specifically, in the 1-shot setting on TACRED, the performance drops from 22.5% to 20.2% when removing type prompt words, indicating that the discriminative prompts are effective for few-shot inference. We conduct further empirical analyses in **Appendix A**, including the analysis of case study, extended ablation study, etc.

## 7 CONCLUSIONS

We first clarify that the over-multitudinous conceptual knowledge contained in PLMs and the incomplete knowledge for the target downstream domain are the essences incurring the defects of state-of-the-art prompt-tuning approaches. To remedy such defects, we propose BayesPrompt to approximate the debiased factual distribution of a downstream domain, and further uniformly sample certain representative features from the approximated distribution to generate the prompts containing the domain discriminative information. Experiments demonstrate the consistent performance superiority of BayesPrompt over baselines.

**Limitations and broader impacts**. The training complexity of BayesPrompt is slightly higher than the baselines, and the distribution approximation possesses the potential for further optimization. Prompt-tuning requires that the target domains and the pre-training domain violate the identically distributed assumption. The scientific question explored by this work only has positive and inspirational impacts on the prompt-tuning community.

ACKNOWLEDGEMENTS

The authors would like to thank the editors and reviewers for their valuable comments. This work is supported by the Fundamental Research Program, China, Grant No. JCKY2022130C020, the National Funding Program for Postdoctoral Researchers, Grant No. GZC20232812, the CAS Project for Young Scientists in Basic Research, Grant No. YSBR-040, the Youth Innovation Promotion Association CAS, No. 2021106, 2022 Special Research Assistant Grant Project, No. E3YD5901, and the China Postdoctoral Science Foundation, No. 2023M743639.

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

# A  FURTHER ANALYSIS

## A.1  CASE STUDY

**Case Study 1:** {"token": ["the", "launcher", "receives", "the", "balls", "through", "a", "similar", "belt", "system", "leading", "up", "the", "neck", "of", "the", "robot", "from", "the", "ground", "."], "h": "name": "launcher", "pos": [1, 2], "t": "name": "system", "pos": [9, 10], "relation": "Instrument-Agency(e2,e1)"}

In the sentence "the launcher receives the balls through a similar belt system leading up the neck of the robot from the ground", the head entity is "launcher", and the tail entity is "system". **KnowPrompt** predicts the relation between the two entities as **"Component-Whole(e2,e1)"**, while **BayesPrompt** predicts the relation between the two entities as **"Instrument-Agency(e2,e1)"**, which aligns with the true label.

**Case Study 2:** {"token": ["from", "banishing", "cold", "and", "flu", "germs", "to", "preventing", "foodborne", "illnesses", ",", "frequent", "hand-washing", "is", "one", "of", "the", "smartest", "preventive", "habits", "you", "can", "adopt", "."], "h": "name": "cold", "pos": [2, 3], "t": "name": "germs", "pos": [5, 6], "relation": "Cause-Effect(e2,e1)"}

In the sentence "from banishing cold and flu germs to preventing foodborne illnesses, frequent hand-washing is one of the smartest preventive habits you can adopt", the head entity is "cold", and the tail entity is "germs". **KnowPrompt** predicts the relation between the two entities as **"Other"**, while **BayesPrompt** predicts the relation between the two entities as **"Cause-Effect(e2,e1)"**, which is in alignment with the true label.

**Case Study 3:** {"token": ["a", "child", "is", "told", "a", "lie", "for", "several", "years", "by", "their", "parents", "before", "he/she", "realizes", "that", "a", "santa", "claus", "does", "not", "exist", "."], "h": "name": "lie", "pos": [5, 6], "t": "name": "parents", "pos": [11, 12], "relation": "Product-Producer(e1,e2)"}

For the sentence "a child is told a lie for several years by their parents before he/she realizes that a santa claus does not exist", the head entity is "lie", and the tail entity is "parents", **KnowPrompt** predicts the relation between the two entities as **"Other"**, while **BayesPrompt** predicts the relation between the two entities as **"Product-Producer(e1,e2)"**, aligning with the true label.

**Conclusion:** We attribute these phenomena to the fact that KnowPrompt only captures the general meanings of entities, whereas BayesPrompt comprehends the genuine context-specific meanings of entities. This can be attributed to the utilization of prompts derived from debiased factual distributions of downstream domains.

## A.2  ANALYSIS OF COMPARATIVE RESULTS BETWEEN GAUSSIAN DISTRIBUTION AND GMM

The innate assumption behind the Gaussian distribution is the sufficient samples, i.e., when the statistical samples are sufficient, the Gaussian distribution well fits the factual distribution of such discrete samples. However, for the few-shot scenario, due to the insufficiency of samples of the target domain, the corresponding distribution of the target domain may not adhere to the typical Gaussian distribution, i.e., the central limit theorem may not well fit the distribution of the target downstream domain. Furthermore, if we strive to use the Gaussian distribution to fit the distribution of the target domain, the derived distribution must have a significant shift from the factual distribution of the target domain. Theoretically, GMM can fit any probability density distribution, so we use Gaussian mixture distribution to approximate the factual distribution of the target domain.

We observe from the empirical results that as the sample size increases, the performance gap between BayesPrompt using GMM and BayesPrompt using the simple Gaussian distribution decreases, which further proves our analysis that when the samples are limited, the distribution of the target domain does not fit the typical Gaussian distribution, while according to the central limit theorem, as the increasing of the sample size, the distribution of the target domain gradually fits the typical Gaussian distribution.

Specifically, for the extreme setting of few-shot learning, e.g., 1-shot, the available data is excessively limited, and the data is insufficient for the fitting of both GMM and the simple Gaussian distribution,

such that the performance of BayesPrompt degenerates with using any candidate distribution. For the relatively limited few-shot data, e.g., 5-shot, BayesPrompt using GMM shows its superiority over the model using the simple Gaussian distribution, which well fits our aforementioned analyses, while, for the relatively sufficient few-shot data, the true distribution of the target domain gradually approaches the Gaussian distribution, such that the performance gap between the two compared model remains decreasing as the increasing of available data size. Eventually, we can still find that BayesPrompt using GMM can consistently outperform BayesPrompt using the simple Gaussian distribution, which indicates that using GMM instead of the typical Gaussian distribution is theoretical and technically solid.

### A.3 Analysis of Performance Improvement in Few-Shot vs. Standard Settings

By comparing Table 2 and Table 3, it can be observed that the performance improvement of BayesPrompt in few-shot scenarios widely surpasses its performance improvement in the standard scenario, i.e., with the full dataset. This observation precisely validates our motivation and the effectiveness of the method.

As we discussed in the Abstract and Introduction sections, the limited and discrete semantic information contained in the training samples from downstream domains can barely support the conventional trainable prompts to acquire sufficient supervision, such that the guidance of the generated prompts is trivial to PLMs. Especially, such a challenge further exacerbates the performance of PLMs in few-shot scenarios. With this motivation, the proposed BayesPrompt aims to learn a prompt that contains relatively sufficient discriminative knowledge for the target downstream domain, which is achieved by proposing the debiased domain abstraction and then generating the prompt. Thus, the empirical observation, i.e., the performance improvement derived by introducing BayesPrompt on few-shot scenarios is more significant than that on standard scenarios, jointly proves the motivation and the effectiveness of the proposed BayesPrompt.

Specifically, in few-shot scenarios, the limited amount of data may hinder PLMs from fully learning the distribution and features of the downstream task data, leading to relatively poorer performance. Therefore, when utilizing prompts obtained from BayesPrompt, which contain domain discriminative information, to guide PLMs in locating knowledge domains relevant to the downstream domain, the deficiencies caused by the limited data are noticeably mitigated. In contrast, in the standard scenario, where the amount of downstream task data is relatively sufficient, PLMs can better grasp the features and distribution of the downstream task data by learning from a more comprehensive dataset. Consequently, the performance improvement brought by BayesPrompt may not be as pronounced in this context. However, the results in standard scenarios further demonstrate the generalizability of our exploration and the effectiveness of BayesPrompt.

### A.4 Extended ablation study and analysis

We further conduct an ablation study on the number of components in the GMM introduced in BayesPrompt, as shown in Table 5. The bolded entries correspond to the selected number of GMM components determined by the number of relation types in the experiment, along with the obtained results. It is observed that, compared to other settings, our choice achieved the best performance. This indicates that our selection of the number of components is appropriate.

The intuition behind the discriminative prompts is that the domain discriminative information is injected into the prompt in BayesPrompt. As shown in Figure 4 (b) and (c), the effectiveness of the type prompt words, i.e., the discriminative prompts proposed by BayesPrompt, is demonstrated. It can be seen that BayesPrompt consistently outperforms the ablation model on all datasets within various experimental settings. Additionally, the performance improvement brought by the discriminative prompts in the few-shot scenario (Figure 4 (b)) is significantly stronger than its performance enhancement in the full dataset (Figure 4 (c)). We attribute this difference to the fact that, in the few-shot scenario, the limited amount of data may hinder PLMs from fully learning the distribution and features of the downstream task data, leading to relatively poorer performance. So, when utilizing the discriminative prompts to guide PLMs in locating knowledge domains relevant to the downstream domain, the deficiencies caused by the limited data are noticeably mitigated. However, in the full dataset, the relatively abundant downstream task data provides a rich knowledge background, resulting in a substantial reduction in the deviation between the knowledge domain located by PLMs

Table 5: F1 scores (%) of BayesPrompt with different numbers of GMM components.

| Dataset | Split | Number of Components in GMM | BayesPrompt |
|---|---|---|---|
| SemEval | K=1 | 9 | 33.7(±3.6) |
| | | **19** | **35.1(±2.9)** |
| | K=5 | 9 | 70.3(±2.8) |
| | | **19** | **71.6(±3.3)** |
| | | 27 | 70.8(±2.7) |
| | K=16 | 9 | 80.8(±1.1) |
| | | **19** | **81.8(±1.2)** |
| | | 27 | 80.8(±1.5) |

and the real downstream knowledge domain. This, in turn, weakens the de-biasing effect of the discriminative prompts on the few-shot dataset. Therefore, the performance improvement brought by the discriminative prompts is more pronounced in the few-shot scenario than in the full dataset, which can effectively prove the validation of our exploration and motivation in the Introduction.

# B  IMPLEMENTATION DETAILS FOR BAYESPROMPT

## B.1  STATISTICAL DETAILS OF DATASETS

We evaluate BayesPrompt on four RE datasets: SemEval 2010 Task 8 (SemEval), TACRED, TACREV, and ReTACRED. SemEval is a popular relation classification dataset that includes 9 relation types with bidirectional labels and an additional "Other" category. TACRED is a widely used dataset for relation classification, obtained through crowd-sourcing, which comprises 42 types of relations, including the "no_relation" label. TACREV is a corrected version of the TACRED dataset, where errors in the original development and test sets were identified and fixed while keeping the training set intact. ReTACRED is another modified version of the TACRED dataset that addresses some of its shortcomings by refactoring its training, development, and test sets and modifying a few relation types. More details of these datasets are shown in Table 6.

Table 6: Statistics of different datasets.

| Dataset | #train | #val | #test | #rel |
|---|---|---|---|---|
| SemEval | 6507 | 1493 | 2717 | 19 |
| TACRED | 68124 | 22631 | 15509 | 42 |
| TACREV | 68124 | 22631 | 15509 | 42 |
| ReTACRED | 58465 | 19584 | 13418 | 40 |

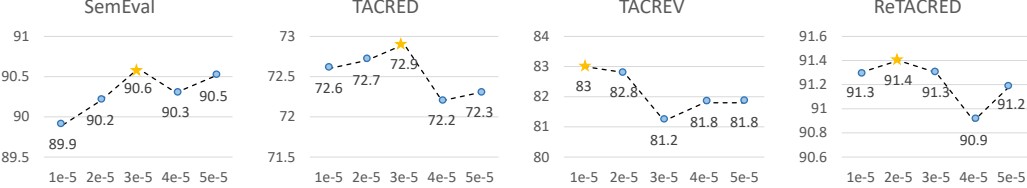

Figure 5: Comparisons of BayesPrompt using different settings of the learning rate.

## B.2  PARAMETERS SETTING IN EXPERIMENTS

The experiments were conducted using Pytorch on 4 Nvidia 3090 GPUs. The learning rate search space of optimization for overall parameters is shown as Figure 5. To form the few-shot training sets, we sample k instances of each relation from the initial training sets. Specifically, we conduct five uniform samplings using a fixed set of seeds and record the average performance and the standard deviation. For all few-shot datasets, we fine-tune our model for 50 epochs with a batch size of 4.

For full datasets, the number of epochs is set to 5, except SemEval, which is set to 10, and the batch size is 16. The best model checkpoint is selected based on the performance of the validation set. To ensure a fair comparison, we use RoBERTa-large for all experiments. We adopt the F1 score as the primary evaluation metric for the experiments on all the aforementioned datasets since the F1 score can comprehensively assess the precision and recall performances.

