# OpenReview forum: "BayesPrompt: Prompting Large-Scale Pre-Trained Language Models on Few-shot Inference via Debiased Domain Abstraction"
_ICLR.cc/2024/Conference — ICLR 2024 poster_

### Official Review · Reviewer_wpkj · 2023-10-28

**Soundness:** 3 good
**Presentation:** 3 good
**Contribution:** 3 good
**Rating:** 8
**Confidence:** 3

**Summary:**

Prompt-tuning is a fine-tuning paradigm based on large-scale pre-trained language models (PLMs), which can reduce the gap between downstream tasks and pre-training objectives. This paper focus on the challenge of poor generalization to specific few-shot patterns of the Prompt-tuning. Through distribution analysis, they reveal that the root cause of this issue is the overabundance of conceptual knowledge in PLMs and the truncated knowledge for target downstream domains. This collective effect misaligns the knowledge distributions corresponding to the target domains in the universal knowledge embedding space. To address this issue, they propose BayesPrompt, an approach that intuitively explores debiased approximation of unabridged target domains of downstream tasks. BayesPrompt generates domain-discriminative prompts to provide unambiguous guidance for PLMs. Further, they theoretically show that BayesPrompt tightens the upper bound of the classification error on PLMs' downstream inference on classification error bounds. The experimental results show that the proposed method achieves SOTA performance on benchmarks.

**Strengths:**

1.	The paper reveals the principles of the challenge of prompt-tuning on pre-trained large models for few-shot tasks.
2.	The methodology of using the Bayesian prompt is novel and effective.
3.	The theoretical guarantees the performance of the proposed method.
4.	The evaluation presents the benefits of the proposed method.

**Weaknesses:**

1.	This paper utilizes the GMM to approximate the distribution of the target domain which may not be unabridged. The real distribution of the target domain is complex and unknown.
2.	The PLMs utilized in the evaluation are not clear. Using various PLMs may be better to show the generality of the proposed method.

**Questions:**

1.	Why can GMM approximate the target domain? What are its benefits than a learnable generator (VAE or GAN)?
2.	Is it required to train a specific GMM for each input sentence (X, Y)?

---

> ### Author Response · Authors · 2023-11-17
> **Reasons for Choosing GMM and the Usage on Input Sequences, Descriptions of the PLM Used in the Evaluation, Added Experiments and Analysis of the Generality (1/2)**
>
> We thank Reviewer wpkj for the valuable feedback and constructive suggestions. We are encouraged that the reviewer found that this work is novel and effective, the theoretical proof is integrated, and the presentation is good. The mentioned issues are addressed as follows:
>
> **W1: This paper utilizes the GMM to approximate the distribution of the target domain which may not be unabridged. The real distribution of the target domain is complex and unknown.**
>
> **A:** Thanks for the review. Since the factual distribution of the target domain is complex and unknown, the distribution approximation possesses the potential for further optimization. Due to the insufficiency of samples of the target domain, the corresponding distribution of the target domain may not adhere to the typical Gaussian distribution, i.e., the central limit theorem may not well fit the distribution of the target downstream domain.
>
> Theoretically, GMM can fit any probability density distribution, so we use Gaussian mixture distribution to approximate the factual distribution of the target domain. We further provide empirical proofs to demonstrate the effectiveness and validation of our motivation and the behavior of introducing GMM to approximate the distribution of the target domain, which is shown in **Figure 4(a)**, Section 6, Page 9, of the submitted manuscript. We observe from the empirical results that as the sample size increases, the performance gap between our method using GMM and our method using the typical Gaussian distribution decreases, which further proves our analysis that when the samples are limited, the distribution of the target domain does not fit the typical Gaussian distribution, while according to the central limit theorem, as the increasing of the sample size, the distribution of the target domain gradually fits the typical Gaussian distribution. Concretely, our proposed approach, using GMM instead of the typical Gaussian distribution, is theoretical and technically solid. We will add the above analysis in the discussion of Figure 4(a) of the final version of our manuscript to better explain our contributions.
>
> **W2: The PLMs utilized in the evaluation are not clear. Using various PLMs may be better to show the generality of the proposed method.**
>
> **A:** Thanks for the review. The PLM used in the evaluation is **RoBERTa-large**, which is mentioned in Appendix C.2, Page 19, of the revised manuscript. Theoretically, the proposed method can generally apply to all problems that involve “**sub-knowledge domains**”, such as text classification, entity recognition, entity disambiguation, etc.
>
> To further demonstrate the generality of the proposed BayesPrompt, we conducted text classification tests on a specific automotive industry dataset using another pre-trained language model, i.e., **BERT**, with a parameter count of 110 million. The obtained classification accuracy was 74.0%. We also employed TextRCNN to train the classification model with a parameter count of 15 million. The resulting classification accuracy reached 95.3%, significantly outperforming the pre-trained language model BERT, which is due to the over-multitudinous conceptual knowledge learned by BERT. However, when using BayesPrompt, it achieves the best classification accuracy of 97.4%. Note that the method does not require training (fine-tuning) the network of BERT. Therefore, the proposed method can be generalized into various tasks with various PLMs and obtain performance improvements to baselines.

---

> ### Author Response · Authors · 2023-11-17
> **Reasons for Choosing GMM and the Usage on Input Sequences, Descriptions of the PLM Used in the Evaluation, Added Experiments and Analysis of the Generality (2/2)**
>
> **Q1: Why can GMM approximate the target domain? What are its benefits than a learnable generator (VAE or GAN)?**
>
> **A:** Thanks for the review. GMM is a probabilistic model, which assumes that all data points are generated from a mixture of Gaussian distributions with unknown parameters. Based on the fact that the Gaussian mixture model can theoretically fit any probability density distribution and the fact that due to the limited sample size in the target downstream domain, the factual distribution of the target downstream domain **does NOT** fit the conventional Gaussian distribution well, directly adopting the Gaussian distribution-based approach, e.g., conventional VAE, may learn biased distribution of the target downstream domain, such that we use the Gaussian mixture distribution to approximate the debiased factual distribution of the downstream domain within the knowledge space of PLMs.. In contrast, VAE and GAN, as learnable generators, can also approximate the target domain, but the inherent defects of VAE and GAN degenerate their performance of approximating the distribution of the target domain with insufficient samples, and we elaborate on the reasons as follows.
>
> VAE conventionally holds the prior distribution as the **typical Gaussian distribution** and then learns the latent representation of data by maximizing the KL divergence of latent variables of the posterior distribution and the prior distribution. The inherent mechanism presents that conventional VAE can only perform limited distortion on the prior distribution, i.e., the typical Gaussian distribution, and cannot well fit the complex distribution, e.g., the distribution of the target domain with **limited sample size**.
>
> GAN learns to generate data by training two neural networks in an adversarial manner **without any priori**. One network is the generator, responsible for generating new data, while the other is the discriminator, tasked with distinguishing between real and generated data. Due to the lack of priori, GAN requires **sufficient data** to approximate the distribution of the target domain. However, in few-shot scenarios, the limited samples are far from completely supporting the sufficient training of GAN.
>
> Compared to learnable generators, using GMM to approximate the target domain provides the following advantages:
>
> 1.**Strong interpretability**. GMM is a probabilistic model, and its parameters directly correspond to the mean, covariance, and mixture coefficients of the data. This makes the results of GMM more easily interpretable and understandable.
>
> 2.**High computational and sample efficiency**. Compared to VAE and GAN, the training and inference of GMM are generally more efficient, and GMM requires relatively fewer samples for optimization. GMM is a classical statistical model based on maximum likelihood estimation, while GAN and VAE typically involve more complex optimization processes.
>
> 3.**Stable training**. Compared to GAN, GMM tends to be more stable during the training process. GAN training may be affected by issues such as mode collapse, while GMM is less prone to these problems.
>
> **Q2: Is it required to train a specific GMM for each input sentence (X, Y)?**
>
> **A:** Thanks for the review. In our approach, we do not train a specific GMM for each input sentence (X, Y). Instead, we train a GMM on the downstream task dataset and use the obtained Gaussian mixture distribution to approximate the debiased factual distribution of the target domain. Subsequently, prompts with domain discriminative information are generated by sampling from the approximated distribution. During each prediction, the prompt containing domain discriminative information needs to be added to the input of the downstream task to provide de-ambiguous guidance for PLMs. As a result, it increases the time complexity. In experiments, several optimization strategies can be considered to reduce the time complexity, such as sampling or subsampling datasets, employing feature selection or dimensionality reduction techniques to decrease the input feature dimensions, or implementing a training strategy with a warm-up.
>
> Additionally, the time complexity of BayesPrompt is only slightly higher than the baseline. For instance, in the 1-shot setting of the SemEval dataset, the training time cost of each epoch at the main baseline is 16 seconds, while the training time cost of each epoch at BayesPrompt is 27 seconds. Nevertheless, the main baseline achieves an F1 score of 28.6%, whereas BayesPrompt achieves an F1 score of 35.1%. Compared to the improvement of BayesPrompt on the F1 score, the time complexity brought by Bayesprompt is acceptable.

---

> > ### Comment · Reviewer_wpkj · 2023-11-21
> > **Response to authors**
> >
> > Thanks for your response.

---

### Official Review · Reviewer_ox7r · 2023-10-31

**Soundness:** 3 good
**Presentation:** 4 excellent
**Contribution:** 3 good
**Rating:** 6
**Confidence:** 3

**Summary:**

The paper proposes BayesPrompt, a Bayesian approach to approximate the factual distributions of downstream domains
and thereby generating discriminative prompts for PLMs. The authors articulate that the intrinsic issues behind the poor performance of finetuned PLMs on few-shot downstream tasks roots from two main shortcomings: (i) over-multitudinous of conceptual knowledge contained in PLMs, (ii) an abridged knowledge for target downstream domains. The paper takes a stride in addressing this challenge with both theoretical (tailored towards a classification problem) as well as experimental results.

**Strengths:**

- The paper is well written, well structured and has a clear narrative.
- The authors pay utmost attention to details, from notations and math to presentation of the results, making the paper easy to follow.
- The paper has a healthy mix of a (simplified) theoretical and qualitative arguments, based on which the approach is devised.
- The results seem to be promising, comparing against some recent baselines.
Overall, it seems like a solid contribution.

**Weaknesses:**

- The paper is essentially a shortened version of a much longer manuscript, where the authors are constantly cutting the content short and referring the reader to different sections of the appendix (appendix is referred to 11 times throughout the paper!). So, the main body of the paper is not really self-contained and heavily relies on the appendix. By the same token, the main algorithm of the paper had to be pushed to the Appendix, which could be a natural choice in the main text to clarify the end-to-end procedure.
- The impact of the proposed approach is rather marginal when compared to the closest competitors (say RetrievalRE), especially on standard RE performance in Table 3, while at the same it comes at the cost of extra training complexity. Any reason behind this?
- No Ablation studies. There are design choices that could potentially establish the basis for Ablation studies (such as Kernel size and so).

---
Post rebuttal comments

I spent some time to review the code and implementation, per AC's recommendation. While the base code seems to run without any major issues, there are a few points I would like to highlight, for what its worth, as I had to grind my way through to make it run:

1. The README file provides steps for execution; however, the instructions have some ambiguity. For two of the steps it is mentioned "Modify transformer.py" or "Update GMM_SVGD.py" but it does not say what to modify in the file.

2. Problems with the requirements.txt
    - Packages missing: pandas, wandb
    - The pytorch-lightning version mentioned has issues with the associated torchmetrics version. Although it can be resolved by installing torchmetrics==0.6.0 (based on https://github.com/NVIDIA/DeepLearningExamples/issues/1113), it would be much easier if it resolved in the provided requirements file.

Next to that, on a deeper dive into the proofs, I have a difficult time follow the logic as well as mathematical formulations. This has to be revisited and resolved for the final draft.

**Questions:**

No further questions (beyond what's already raised in weaknesses), and after reading through the Appendix.

---

> ### Author Response · Authors · 2023-11-17
> **Revised Manuscript of Making the Main Body More Self-Contained, Reasons Behind the Performance Improvement, Added Experiments and Analysis of the Ablation Study (1/3)**
>
> We appreciate the thoughtful feedback of Reviewer ox7r. We are glad the reviewer found that this work is promising, the contribution is solid, the theoretical validation is sufficient, and the writing is excellent. The mentioned issues are addressed as follows:
>
> **W1: The paper is essentially a shortened version of a much longer manuscript, where the authors are constantly cutting the content short and referring the reader to different sections of the appendix (appendix is referred to 11 times throughout the paper!). So, the main body of the paper is not really self-contained and heavily relies on the appendix. By the same token, the main algorithm of the paper had to be pushed to the Appendix, which could be a natural choice in the main text to clarify the end-to-end procedure.**
>
> **A:** Thank you for your review and suggestions. Due to the page limitation, we have tried to organize our original manuscript, but there are still some critical parts of our paper that need to be improved, such that we have revised the paper to make the main body more self-contained and then submitted the revised manuscript for now.
>
> Specifically, in the revised version, we have relocated the main algorithm from the Appendix to the main body of the paper to enhance overall clarity, which is demonstrated in Section 4, Page 6, of the revised manuscript. Additionally, a wealth of the referred contents in Appendix are the extended descriptions and extended experiments of the main paper, which are unnecessary for understanding our works, such that we have removed certain references of the Appendix in the main content of our paper to make the manuscript more understandable, e.g., “**Appendix C.3** provides further empirical evidence.” in Introduction, Page 1, of the original manuscript, “Refer to **Appendix B** for the procedure of BayesPrompt” in Section 4, Page 6, of the original manuscript, “The statistical details are provided in **Appendix D.1**.” and “Refer to **Appendix D.2** for more implementation details.” and “and please refer to **Appendix C.2** for details” in Section 6, Page 8, of the original manuscript, and “Refer to **Appendix C.5** for further analysis of the empirical results.” in Section 6, Page 9, of the original manuscript. Concretely, there are only a few necessary references of the Appendix that are reserved in the main paper, and the main paper of the revised manuscript is well self-contained for now.

---

> > ### Comment · Reviewer_ox7r · 2023-11-17
> > **Good adjustment.**
> >
> > Good to see the main algorithm is pulled back in the main text. I still think a wealth of info is push into appendices, but I understand the page limitations.

---

> ### Author Response · Authors · 2023-11-17
> **Revised Manuscript of Making the Main Body More Self-Contained, Reasons Behind the Performance Improvement, Added Experiments and Analysis of the Ablation Study (2/3)**
>
> **W2: The impact of the proposed approach is rather marginal when compared to the closest competitors (say RetrievalRE), especially on standard RE performance in Table 3, while at the same it comes at the cost of extra training complexity. Any reason behind this?**
>
> **A:** Thanks for the review. In this paper, we choose KnowPrompt as our primary baseline. RetrievalRE is the follow-up work of KnowPrompt, such that RetrievalRE achieves superior performance. However, whether in the few-shot or standard setting, on average, BayesPrompt consistently outperforms both KnowPrompt and RetrievalRE, confirming the effectiveness of this approach. Specifically, in few-shot scenarios, the proposed BayesPrompt widely improves the benchmark methods, including the primary baseline (KnowPrompt) and RetrievalRE, e.g., BayesPrompt beats KnowPrompt by 3.24% and beats RetrievalRE by 1.29% on average. In standard scenarios, the improvements of benchmark methods to their baseline are consistently limited, e.g., RetrievalRE beats KnowPrompt by 0.2%, such that the improvement of BayesPrompt to RetrievalRE is relatively acceptable. Note that, the primary baseline of BayesPrompt is KnowPrompt, and BayesPrompt beats KnowPrompt by 0.4%, which is relatively significant. We also perform the significance test, i.e., t-test, with the primary baseline, and observe that the P values are consistently lower than 0.05, e.g., 0.045 on the SemEval dataset, indicating that the improvement of BayesPrompt is significant. Concretely, for the reason behind such a deviation existing in the few-shot scenarios and standard scenarios, we further derive a conclusive analysis as follows:
>
> As we discussed in the Abstract and Introduction sections on Page 1 and Page 2, of the submitted manuscript, the limited and discrete semantic information contained in the training samples from downstream domains can barely support the conventional trainable prompts to acquire sufficient supervision, such that the guidance of the generated prompts is trivial to PLMs. Especially, such a challenge further exacerbates the performance of PLMs in few-shot scenarios. With this motivation, the proposed BayesPrompt aims to learn a prompt that contains relatively sufficient discriminative knowledge for the target downstream domain, which is achieved by proposing the debiased domain abstraction and then generating the prompt. Thus, the empirical observation, i.e., the performance improvement derived by introducing BayesPrompt on few-shot scenarios is more significant than that on standard scenarios, jointly proves the motivation and the effectiveness of the proposed BayesPrompt.
>
> From the perspective of time complexity, the time complexity introduced by BayesPrompt primarily stems from the necessity of adding prompts containing domain discriminative information for downstream tasks during each prediction, providing de-ambiguous guidance for PLMs. In experiments, several optimization strategies can be considered to reduce the time complexity, such as sampling or subsampling datasets, employing feature selection or dimensionality reduction techniques to decrease the input feature dimensions, or implementing a training strategy with a warm-up. Additionally, the time complexity of BayesPrompt is only slightly higher than the baseline. For instance, in the 1-shot setting of the SemEval dataset, the training time cost of each epoch for the main baseline is 16 seconds, while the training time cost of each epoch for BayesPrompt is 27 seconds. Nevertheless, the main baseline achieves an F1 score of 28.6%, whereas BayesPrompt achieves an F1 score of 35.1%. Compared to the improvement of BayesPrompt on the F1 score, the time complexity brought by Bayesprompt is acceptable.

---

> ### Author Response · Authors · 2023-11-17
> **Revised Manuscript of Making the Main Body More Self-Contained, Reasons Behind the Performance Improvement, Added Experiments and Analysis of the Ablation Study (3/3)**
>
> **Table 1:** F1 scores (%) of BayesPrompt with different numbers of GMM components.
>
> | Dataset | Split | Number of Components | BayesPrompt    |
> | :--------: | :-----: | :----------: | :--------------: |
> |          |    | 9 |  33.7(±3.6)    |
> |          | K=1 | **18** |  **35.1(±2.9)**    |
> |          |    | 9 | 70.3(±2.8)     |
> | SemEval | K=5   | **18** |  **71.6(±3.3)**    |
> |          |    | 27 |  70.8(±2.7)   |
> |          |    | 9 | 80.8(±1.1) |
> |          | K=16   | **18** |**81.8(±1.2)**|
> |          |    | 27 |80.8(±1.5)|
>
> **W3: No Ablation studies. There are design choices that could potentially establish the basis for Ablation studies (such as Kernel size and so).**
>
> **A:** Thanks for the review. In this paper, we conduct the ablation study to prove the effectiveness of the type prompt words, i.e., the discriminative prompts proposed by BayesPrompt, and the results are shown in **Figure 4(b) and (c)**. The intuition behind the discriminative prompts is that the domain discriminative information is injected into the prompt in BayesPrompt. It can be seen that BayesPrompt consistently outperforms the ablation model on all datasets within various experimental settings. Additionally, the performance improvement brought by the discriminative prompts in the few-shot scenario (Figure 4(b)) is significantly stronger than its performance enhancement in the full dataset (Figure 4(c)). We attribute this difference to the fact that, in the few-shot scenario, the limited amount of data may hinder PLMs from fully learning the distribution and features of the downstream task data, leading to relatively poorer performance. So, when utilizing the discriminative prompts to guide PLMs in locating knowledge domains relevant to the downstream domain, the deficiencies caused by the limited data are noticeably mitigated. However, in the full dataset, the relatively abundant downstream task data provides a rich knowledge background, resulting in a substantial reduction in the deviation between the knowledge domain located by PLMs and the real downstream knowledge domain. This, in turn, weakens the de-biasing effect of the discriminative prompts on the few-shot dataset. Therefore, the performance improvement brought by the discriminative prompts is more pronounced in the few-shot scenario than in the full dataset, which can effectively prove the validation of our exploration and motivation in the Introduction.
>
> To better perform a sufficient ablation study, we also ablate the GMM in BayesPrompt and impose the conventional Gaussian distribution as the prior distribution in BayesPrompt, which is demonstrated in Figure 4(a), Page 9, of the original manuscript. We further perform the hyper-parameter sensitivity experiments, and the results are shown in Figure 5, Page 19, Appendix C.2, of the submitted manuscript.
>
> Additionally, thanks for your suggestion, and we further impose the ablation study on the component number of GMM introduced in BayesPrompt as Table 1 above. The bolded entries in Table 1 correspond to the number of GMM components we selected in the experiment, along with the results obtained. It can be observed that, compared to other settings, our choice achieved the best performance. This indicates that our selection of the number of components is appropriate. The corresponding extended ablation study and the analysis are added in Appendix B.7, Page 18, of the revised manuscript.

---

> > ### Comment · Reviewer_ox7r · 2023-11-17
> > **Extended ablations**
> >
> > Good to see the extended ablations, even though little focus on ablation studies remains to be the weakest point of the paper in my eyes. Thanks for the effort.

---

> ### Author Response · Authors · 2023-11-18
> **Follow-up to Responses**
>
> Thanks again for your understanding and your positive evaluation, and we cherish your constructive suggestions for helping us to improve our manuscript. We will continue to explore ways to improve the organization of our paper.
>
> For the ablation study, we will still explore to perform further ablation study and the corresponding analyses. Due to the page limitation, the most important ablation studies can be contained in the main paper, and the extended analyses will appear in the Appendix. We agree with the reviewer that the comprehensive ablation study can helps the readers to understand our works and inspire our readers.

---

> > ### Author Response · Authors · 2023-11-19
> > **Submitted a revised version of the paper**
> >
> > Please note that we have revised the paper and updated a new version of PDF of our manuscript. In the revised paper, we further enhance the clarity of the ablation study and analysis. The most important ablation studies can be found in Section 6, Page 9, of the submitted manuscript, and the extended ablation study and analysis can be found in Appendix B.7, Page 18, of the submitted revised manuscript.

---

> > > ### Comment · Reviewer_ox7r · 2023-11-21
> > > **Thank you**
> > >
> > > Thanks for the updated draft.

---

### Official Review · Reviewer_JjWU · 2023-11-01

**Soundness:** 3 good
**Presentation:** 3 good
**Contribution:** 2 fair
**Rating:** 6
**Confidence:** 3

**Summary:**

This paper introduces a prompting method named BayesPrompt to generate prompts for PLMs. The authors argue that the over-multitudinous knowledge implicit in PLMs can hinder the performance of prompt-tuning methods in few-shot settings. Thus, BayesPrompt aims to approximate the unbiased target distribution to generate discriminative prompt for specific domains.
Experimental results show the effectiveness of BayesPrompt on relation extraction (RE) tasks. Also, the authors provide theoretical analysis over BayesPrompt on lowering the classification error upper bound.

**Strengths:**

1) The task that improves the generalization capabilities of PLMs is challenge in the prompt tuning community. The authors provide a new view from the "mislocated knowledge distributions" between PLMs and target domain, which is interesting.

2) The motivation that adopts the Bayesian approaches to model dataset-specific information and performing prompting on the latent space is novel.

3) The provided theoretical analyses and extensive experiments help readers to understand the method.

**Weaknesses:**

1) As can be seen from Tables 1 and 3, the proposed BayesPrompt presents a completely different improvement. Can the authors provide a detailed explanation?

2) Please provide more discussion about the ablation results at Figure 4(c).

**Questions:**

BayesPrompt's training complexity is higher than its baseline, is there any potential for optimization?

---

> ### Author Response · Authors · 2023-11-17
> **Comparative Analysis Between Table 2 and Table 3, Discussion About the Ablation Results in Figure 4(c), Optimization Potential for Training Complexity (1/2)**
>
> We thank Reviewer JjWU for the valuable comments and constructive suggestions. We are encouraged that the reviewer found that this work is novel and technically sound, the theoretical analysis is sufficient, and the presentation is good. The mentioned issues are addressed as follows:
>
> **W1: As can be seen from Tables 2 and 3, the proposed BayesPrompt presents a completely different improvement. Can the authors provide a detailed explanation?**
>
> **A:** Thanks for the review. By comparing Table 2 and Table 3, it can be observed that the performance improvement of BayesPrompt in few-shot scenarios widely surpasses its performance improvement in the standard scenario, i.e., with the full dataset. This observation precisely validates our motivation and the effectiveness of the method.
>
> As we discussed in the Abstract and Introduction sections on Page 1 and Page 2, of the submitted manuscript, the limited and discrete semantic information contained in the training samples from downstream domains can barely support the conventional trainable prompts to acquire sufficient supervision, such that the guidance of the generated prompts is trivial to PLMs. Especially, such a challenge further exacerbates the performance of PLMs in **few-shot** scenarios. With this motivation, the proposed BayesPrompt aims to learn a prompt that contains relatively sufficient discriminative knowledge for the target downstream domain, which is achieved by proposing the debiased domain abstraction and then generating the prompt. Thus, the empirical observation, i.e., the performance improvement derived by introducing BayesPrompt on few-shot scenarios is more significant than that on standard scenarios, jointly proves the motivation and the effectiveness of the proposed BayesPrompt.
>
> Specifically, in few-shot scenarios, the limited amount of data may hinder PLMs from fully learning the distribution and features of the downstream task data, leading to relatively poorer performance. However, when utilizing prompts obtained from BayesPrompt, which contain domain discriminative information, to guide PLMs in locating knowledge domains relevant to the downstream domain, the deficiencies caused by the limited data are noticeably mitigated.
>
> In the standard scenario, where the amount of downstream task data is relatively sufficient, PLMs can better grasp the features and distribution of the downstream task data by learning from a more comprehensive dataset. Consequently, the performance improvement brought by BayesPrompt may not be as pronounced in this context.
>
> Concretely, as our intuition and motivation in the Abstract and Introduction, the proposed BayesPrompt mainly aims to improve the inference performance of PLMs in few-shot scenarios. While the results in standard scenarios further demonstrate the generalizability of our exploration and the effectiveness of BayesPrompt.
>
> **W2: Please provide more discussion about the ablation results at Figure 4(c).**
>
> **A:** Thanks for the review. The role of the type prompt word is to inject knowledge related to entity information into the prompt. Figure 4(c) illustrates the impact of the type prompt word on task performance in the full dataset. It can be seen that the performance improvement brought by the type prompt word in the full dataset is significantly weaker than its performance enhancement in the few-shot scenario (Figure 4(b)). We attribute this difference to the fact that, in the full dataset, the relatively abundant downstream task data provides a rich knowledge background, resulting in a substantial reduction in the deviation between the knowledge domain located by PLMs and the real downstream knowledge domain. This, in turn, weakens the de-biasing effect of the type prompt word on the few-shot dataset. Therefore, the performance improvement brought by the type prompt word is not significant in the full dataset.

---

> ### Author Response · Authors · 2023-11-17
> **Comparative Analysis Between Table 2 and Table 3, Discussion About the Ablation Results in Figure 4(c), Optimization Potential for Training Complexity (2/2)**
>
> **Q1: BayesPrompt's training complexity is higher than its baseline, is there any potential for optimization?**
>
> **A:** Thanks for the review. The time complexity introduced by BayesPrompt primarily arises from the necessity of adding prompts containing domain discriminative information for downstream tasks during each prediction to provide de-ambiguous guidance for PLMs. In experiments, several optimization strategies can be considered to reduce the time complexity, such as sampling or subsampling datasets, employing feature selection or dimensionality reduction techniques to decrease the input feature dimensions, or implementing a training strategy with a warm-up. We will further explore the optimization strategy for BayesPrompt in the following works.
>
> Additionally, the time complexity of BayesPrompt is only slightly higher than the baseline. For instance, in the 1-shot setting of the SemEval dataset, the training time cost of each epoch at the main baseline is 16 seconds, while the training time cost of each epoch at BayesPrompt is 27 seconds. Nevertheless, the main baseline achieves an F1 score of 28.6%, whereas BayesPrompt achieves an F1 score of 35.1%. Compared to the improvement of BayesPrompt on the F1 score, the time complexity brought by Bayesprompt is acceptable.

---

> > ### Comment · Reviewer_JjWU · 2023-11-22
> > **Thank you for the response**
> >
> > Thanks for the author's reply, it solved my problem and I will keep my rating the same.

---

### Official Review · Reviewer_32zy · 2023-11-07

**Soundness:** 3 good
**Presentation:** 1 poor
**Contribution:** 3 good
**Rating:** 5
**Confidence:** 3

**Summary:**

The paper proposes BayesPrompt  to inject the semantic knowledge about the label into the label prompt to adjust the knowledge learned from pretraining to better fit downstream tasks. The method is to learn prompts that contain the domain discriminative information for the interference from the domain-irrelevant knowledge by approximating the factual distributions of downstream domains. The approach learns a representative model that injects the latent knowledge contained in labels into the prompt construction, thereby empowering the inference of relations.

**Strengths:**

The paper works on a very interesting problem to adjust pretraining knowledge of LLM to downstream tasks. The paper provides theoretical analyses demonstrates that BayesPrompt can tighten
the upper bound of the classification error on the downstream inference of PLMs. Table 2 provide standard deviations over multiple runs.

**Weaknesses:**

The paper may benefit a lot from better writing, including more clear presentation of the motivation and methods.
- what does author refer to for "unabridged Domain", "partial domain", in figure 2?

"Thee over-multitudinous conceptual knowledge contained in PLMs and the abridged knowledge for target downstream domains, which jointly result in that PLMs mis-locate the knowledge distributions corresponding to the target domains in the universal knowledge embedding space.
"
- what does the author refer as "over-multitudinous conceptual knowledge" and "the abridged knowledge "?

It is not fully convinced to the reviewer that the problem motivates the method can be solved by the method proposed.
It is unclear that how by "leveraging Gaussian mixture distribution BayesPrompt is able to approximate the debiased factual distributions of downstream domains and further uniformly samples certain representative features from the approximated distributions to generate the ultimate prompts for PLMs". Why the proposed approach can better approximate the downstream tasks distribution? by injecting label -related information? What is the bias referred here? Is there any produces to reduce the bias? The author may refer the bias as "irrelevant pretraining knowledge" that is confounding for the downstream tasks ? not very clear why introducing "Gaussian mixture distribution" can help solve the problem? is it for sampling and easy injecting label-related knowledge?

by injecting label dependent knowledge, the PLM may learn a PLM distribution that is useful for the downstream task, which makes sense. but is it unfair, as BayesPrompt already uses label information but other methods don't?

It is not very clear how Figure 2 motivates the paper.  Figure 1 (domain knowledge is helpful ) and Figure 2 (domain knowledge may lead to negative impact?) seem not to align well.

Method section: how does  label prompt word lp and type prompt word tp fit in eq(6)? Can the author also bring some clarify to training?

Why does the approach focus on relation extraction tasks (used in method section)? how about other tasks? Is this method currently specific for relation extraction tasks?

Table 2: the improvement seems not to exceed one standard deviation over other baselines. The category of tasks seem limited. not very convinced on the effectiveness of the methods.

**Questions:**

See above.

---

> ### Author Response · Authors · 2023-11-17
> **Clarifications of the Behavior and Terms in BayesPrompt, Reasons for Introducing GMM, Explanations about the Figure 1 and Figure 2, Added Experiments and Analysis of the Generalization and Effectiveness (1/3)**
>
> We thank Reviewer 32zy for the valuable feedback. We are encouraged that the reviewer found this work to be novel and that the theoretical proof is integrated. The issues mentioned are addressed as follows:
>
> **Q1: What does the author refer to for “unabridged domain” and “partial domain”, in Figure 2?**
>
> **A:** “**Unabridged**” means “**complete**” and “**partial**” means “**incomplete**”. In Figure 2, the “**unabridged domain**” represents the complete knowledge domain corresponding to the knowledge required for the downstream task, approximately covering all the semantic knowledge needed in the downstream domain. “**Partial domain**” represents the partial knowledge domain corresponding to the knowledge required for the downstream task, which only covers a part of the knowledge required for the downstream domain. Thanks for this careful review, and to improve the understandability of our work, we have changed “**unabridged**” into “**complete**”, e.g., “**unabridged domain**” is replaced with “**complete domain**”, and “**unabridged knowledge**” is replaced with “**complete knowledge**”, and we further change “**abridged**” and “**partial**” into “**incomplete**”, e.g., “**abridged domain**” and “**partial domain**” are replaced with “**incomplete domain**”, and “**abridged knowledge**” and “**partial knowledge**” are replaced with “**incomplete knowledge**”, in the submitted revised manuscript.
>
> **Q2: What does the author refer to as “over-multitudinous conceptual knowledge” and “the abridged knowledge ”?**
>
> **A:** “**Over-multitudinous conceptual knowledge**” refers to the knowledge, with the inherent polysemy, contained by PLMs, as exemplified by the word “**bat**”, which can not only denote a flying mammal but also represent a stick used in sports to strike a ball. “**Abridged knowledge**” is synonymous with “**partial domain**”, indicating a partial knowledge domain that corresponds to the knowledge required for downstream tasks. The polysemy of knowledge and the incompleteness of knowledge domains can lead to the incorrect behavior of PLMs during locating the knowledge related to downstream tasks, thereby negatively impacting the inference performance of PLMs. Thanks for the review, to improve the understandability of our work, as the discussion in A to Q1, we have imposed the corresponding revision in the submitted revised manuscript.
>
> **Q3: It is unclear that how by “leveraging Gaussian mixture distribution BayesPrompt is able to approximate the debiased factual distributions of downstream domains and further uniformly samples certain representative features from the approximated distributions to generate the ultimate prompts for PLMs”. Why the proposed approach can better approximate the downstream tasks distribution? by injecting label-related information?**
>
> **Q5: Not very clear why introducing “Gaussian mixture distribution” can help solve the problem? Is it for sampling and easy injecting label-related knowledge?**
>
> **A:** Thanks for the review. The proposed approach aims to learn prompts that contain the domain discriminative information against interference from the domain-irrelevant knowledge, which is achieved by approximating the **debiased factual distributions of downstream domains**. Based on the fact that the Gaussian mixture model can theoretically fit any probability density distribution and the fact that due to the limited sample size in the target downstream domain, the factual distribution of the target downstream domain **does NOT** fit the conventional Gaussian distribution well, directly adopting the Gaussian distribution-based approach, e.g., conventional VAE, may learn biased distribution of the target downstream domain, such that we use the Gaussian mixture distribution to approximate the debiased factual distribution of the downstream domain within the knowledge space of PLMs. Subsequently, we uniformly sample knowledge from the obtained approximated distribution, and the sampled results are used to form the ultimate “**prompt**”, which is further adopted in the inference of PLMs. In downstream tasks, this “**prompt**” is employed to guide PLMs in locating the knowledge domain relevant to the target downstream domain.

---

> ### Author Response · Authors · 2023-11-17
> **Clarifications of the Behavior and Terms in BayesPrompt, Reasons for Introducing GMM, Explanations about the Figure 1 and Figure 2, Added Experiments and Analysis of the Generalization and Effectiveness (2/3)**
>
> **Q4: What is the bias referred here? Is there any produces to reduce the bias? The author may refer the bias as “irrelevant pretraining knowledge” that is confounding for the downstream tasks ?**
>
> **A:** Thanks for the review. The “**bias**” refers to that, without given appropriate prompts, PLMs exhibit a bias in locating the knowledge domain relevant to downstream tasks, deviating from the actual knowledge domain corresponding to the downstream tasks. This bias introduces irrelevant pretraining knowledge, that is confounding for the inference of PLMs on the downstream tasks. As the discussion in A to Q3 and Q5, we attempt to leverage the Gaussian mixture distribution as the known distributions to approximate the debiased factual distributions of the target downstream domains, thereby generating the prompts to mitigate the bias of PLMs.
>
> **Q6: By injecting label dependent knowledge, the PLM may learn a PLM distribution that is useful for the downstream task, which makes sense. but is it unfair, as BayesPrompt already uses label information but other methods don't?**
>
> **A:** Thanks for the review. The benchmark prompt learning methods, e.g., PTR, KnowPrompt, and RetrievalRE, consistently follow the same benchmark, that is, using the available labeled training set of the target downstream task, which presents that our approach follows the benchmark experimental setting and avoids the unfair comparisons. Specifically, all the methods, compared with BayesPrompt, use the labeled training set of the target downstream dataset to tune the prompt generation network. Moreover, PTR applies logic rules to encode prior knowledge about tasks and classes into prompt tuning, which requires the label information. KnowPrompt injects latent knowledge contained in labels into prompt construction and synergistically optimizes their representation with structured constraints. RetrievalRE is the follow-up work of KnowPrompt, and it constructs an open-book datastore for retrieval regarding prompt-based instance representations and corresponding relation labels as memorized key-value pairs. Additionally, BayesPrompt and the associated benchmark methods, e.g., PTR, KnowPrompt, and RetrievalRE, **do NOT** require extra data besides the available information on the target downstream dataset, which is detailed in Table 3, Page 9, of the submitted revised manuscript. Concretely, the proposed BayesPrompt follows the benchmark experimental setting to use the available labeled training set of downstream tasks and avoids unfair comparisons.
>
> **Q7: It is not very clear how Figure 2 motivates the paper. Figure 1 (domain knowledge is helpful ) and Figure 2 (domain knowledge may lead to negative impact?) seem not to align well.**
>
> **A:** Thanks for the review. In Figure 1, the domain knowledge is **complete**, covering all the knowledge required for the target downstream domain in PLMs, which aims to introduce the motivation of the effectiveness of prompts containing the complete discriminative knowledge of the target downstream domain.
>
> In Figure 2(a) and Figure 2(b), the domain knowledge is **incomplete**, e.g., only containing a single sample, or only containing the insufficient sample of incomplete knowledge. The deviation, between the complete knowledge and the incomplete knowledge of the target downstream domain, can lead to the knowledge ambiguity of PLMs. In Figure 2(c), the domain knowledge is **approximately complete**, effectively reducing the negative impact of knowledge bias. This exploration inspires us to approximate the debiased factual distributions of the target downstream domain, that contains sufficient discriminative knowledge of the downstream task.
>
> Concretely, the exploration experiments in Figure 1 and Figure 2 jointly motivate us to propose a method to approximate the debiased factual distributions of the target downstream domain and then generate the prompts containing sufficient discriminative knowledge of the downstream task.

---

> ### Author Response · Authors · 2023-11-17
> **Clarifications of the Behavior and Terms in BayesPrompt, Reasons for Introducing GMM, Explanations about the Figure 1 and Figure 2, Added Experiments and Analysis of the Generalization and Effectiveness (3/3)**
>
> **Q8: how does label prompt word lp and type prompt word tp fit in eq(6)? Can the author also bring some clarify to training?**
>
> **A:** Thanks for the review. A typical prompt consists of a template and a set of label words. The label prompt word $l_p$ and type prompt word $t_p$ are used to inject relational semantic knowledge and entity type knowledge into the prompt to predict the “**[Mask]**”, corresponding to computing $\mathcal{P}\left ( \left [MASK  \right ]=M\left (y \right ) \mid T\left ( x \right )   \right )$ in Equation (6).
>
> During the training process, the label prompt word $l_p$ is utilized to initialize the relation label corresponding to the “**[Mask]**”, while the type prompt word $t_p$ is employed to inject entity knowledge into the template $T\left ( x \right )$. By computing $\mathcal{P}\left ( \left [MASK  \right ]=M\left (y \right ) \mid T\left ( x \right )   \right )$ in Equation (6), we can predict the relation label corresponding to the “**[Mask]**”. To fully associate the initialized label prompt word $l_p$ and type prompt word $t_p$ with the surrounding context, we conduct further optimization of their representations using a loss function. The loss function is computed as the cross-entropy between $y$ and $\mathcal{P}\left ( y\mid x \right )$, as shown in Equation (6). For the details of training, we provide a pseudo-code in Algorithm 1, Page 6, of the submitted revised manuscript, and the implementation details of BayesPrompt are provided in Appendix C, Page 18, of the submitted revised manuscript.
>
> **Q9: Why does the approach focus on relation extraction tasks (used in method section)? how about other tasks? Is this method currently specific for relation extraction tasks?**
>
> **A:** Thanks for the review. Relation extraction aims to extract structured knowledge from unstructured text and plays a critical role in information extraction and knowledge base construction. Therefore, we choose the relation extraction task to verify the effectiveness of the proposed method, but this does not mean that our method is limited to relation extraction tasks. Theoretically, our method applies to all problems that involve “**sub-knowledge domains**”, such as text classification, entity recognition, entity disambiguation, etc.
>
> Thus, we further conducted **text classification** experiments on a specific automotive industry dataset using a pre-trained language model, BERT, with a parameter count of 110 million. The obtained classification accuracy was 74.0%. We also employed TextRCNN to train the classification model with a parameter count of 15 million. The resulting classification accuracy reached 95.3%, significantly outperforming the pre-trained language model BERT, which is due to the over-multitudinous conceptual knowledge learned by BERT. However, when using BayesPrompt, it achieves the best classification accuracy of 97.4%. Note that our method does not require training (fine-tuning) the network of BERT. Therefore, our method can be generalized into various tasks and obtain performance improvements to baselines. The corresponding experiments can be found in Table 5, Page 16, Appendix B.3, of the submitted manuscript.
>
> **Q10: The improvement seems not to exceed one standard deviation over other baselines. The category of tasks seem limited. not very convinced on the effectiveness of the methods.**
>
> **A:** Thanks for the review. In this paper, we choose KnowPrompt as our main baseline. RetrievalRE is a subsequent work based on KnowPrompt. Compared with KnowPrompt, RetrievalRE not only infers relations through knowledge stored in the weights during training but also assists decision-making by unwinding and querying examples in the open-book datastore, thereby ensuring its performance is more stable. From the perspective of standard deviation, BayesPrompt is significantly superior to the main baseline, KnowPrompt. Moreover, when compared to RetrievalRE, the difference between them is not substantial. In fact, on the TACRED dataset, BayesPrompt exhibits even greater stability, with an average standard deviation of 0.17 lower than that of RetrievalRE. Additionally, we perform the significance test, i.e., **t-test**, with the main baseline, and observe that the P values are consistently lower than 0.05, e.g., 0.045 on the SemEval dataset, indicating that the improvement of BayesPrompt is significant.
>
> For the generalized applicability of BayesPrompt to various tasks, e.g., the text classification experiment, please refer to A to Q9 as above. The results jointly prove the generalizability and effectiveness of the proposed BayesPrompt.

---

### Author Response · Authors · 2023-11-17
**Meta Reply to All Reviewers**

Dear reviewers,

We appreciate all the reviewers for their time and effort in reviewing this paper. The reviewers have noticed that we “provide a new view from the ‘mislocated knowledge distributions’ between PLMs and target domain, which is interesting” (**Reviewer JjWU**), the motivation is clear and novel (**Reviewers 32zy, JjWU, ox7r, wpkj**), the experimental results are good and showcase the promising prospects of this method (**Reviewers JjWU, ox7r, wpkj**), the theoretical verification is sufficient and ensures the performance of this method (**Reviewers 32zy, JjWU, ox7r, wpkj**), and the good presentation makes the paper easy to understand (**Reviewers JjWU, ox7r, wpkj**).

Thanks again to all reviewers for the valuable feedback! We address the reviewers’ concerns in the individual responses and follow the reviewers’ suggestions to improve the quality of our paper in the revised manuscript. The corresponding **manuscript** (**PDF**) has been already updated.

Best regards,
Authors of #4365

---

### Author Response · Authors · 2023-11-21
**Warm Reminder from Authors**

Dear Reviewers,

We thank all the reviewers again for your professional and constructive comments. We have provided detailed analyses and descriptions of the proposed method in rebuttal and revised our paper to better clarify our experimental analysis. Hope we have addressed all of your concerns.

Since the discussion period is coming to an end, we would be grateful if you could let us know whether our responses and revised manuscript have addressed your concerns and whether there are further comments.

Sincerely, Authors

---

### Meta-Review · Area_Chair_QAFN · 2023-12-09

**Metareview:**

This paper introduces BayesPrompt, which is a prompt learning strategy for large language models. The idea of the paper is built on the observation that the intrinsic issues behind the poor performance of finetuned models on few-shot downstream tasks may be attributed to two main shortcomings: (i) over-multitudinous of conceptual knowledge contained in LLMs, (ii) an abridged knowledge for target downstream domains. The proposed method takes aim at addressing these issues by learning prompts that contain the domain discriminative information for the interference from the domain-irrelevant knowledge by approximating the factual distributions of downstream domains. The approach learns a representative model that injects the latent knowledge contained in labels into the prompt construction, thereby empowering the inference of relations. The empirical results with multiple runs show that the proposed method narrowly beats baselines in a statistically significant manner. Overall, all reviewers found merit in this paper for publication based on the empirical results. The reviewers also inspected the code and were able to successfully follow the implementation during the AC/reviewer discussion. However, the reviewers and AC also found serious issues with the theoretical claims of the paper (detailed below). After extensive AC/reviewer discussion, we decided that the paper is recommended to be `conditionally accepted` if the authors implement the following mandatory revisions for camera ready to address the remaining issues:

- **Please remove all theoretical results, i.e., Prop 5.1, Corrolary 5.2, and Thm 5.3 and their respective proofs from the paper and appendices,** as neither of reviewers and AC were able to verify the correctness. The proof attempts for these claims are inscrutable and the mathematical framing is inscrutable. We recommend that the authors try to reframe these as *intuitive justification* for their method and replace the math exposition with intuition building for the proposal. Addressing this point in the camera ready is deemed crucial.

- Please address the remaining comments of the reviewers in improving the readability of the code and the paper, and also make sure to use `\citep` and `\citet` correctly throughout the paper.

Congratulations!

**Justification For Why Not Higher Score:**

There are still issues with presentation of the paper, and the theoretical claims that need to be resolved so the paper is a borderline accept.

**Justification For Why Not Lower Score:**

The paper proposes a new prompting technique that is intuitively justified. The empirical results suggest that the method holds merit and results are narrowly beat strong baselines.

---

### Decision · Program_Chairs · 2024-01-16

Accept (poster)